# Occurrence rate of ultra-low frequency waves in the foreshock of Mercury increases with heliocentric distance

N. Romanelli [1,2,3,4✉] & G. A. DiBraccio[2]

Studies of Mercury's foreshock have analyzed in detail the properties of ultra-low frequency waves. However, an open question remains in regards to understanding favorable conditions for these planetary foreshocks waves. Here, we report that 0.05–0.41 Hz quasi-monochromatic waves are mostly present under quasi-radial and relatively low intensity Interplanetary Magnetic Field, based on 17 Mercury years of MESSENGER Magnetometer data. These conditions are consistent with larger foreshock size and reflection of solar wind protons, their most likely source. Consequently, we find that the wave occurrence rate increases with Mercury's heliocentric distance. Detection of these waves throughout Mercury's highly eccentric orbit suggests the conditions for backstreaming protons are potentially present for all of Mercury's heliocentric distances, despite the relatively low solar wind Alfvén Mach number regime. These results are relevant for planetary magnetospheres throughout the solar system, and the magnetospheres of exoplanets, and provide knowledge of particle acceleration mechanisms occurring inside foreshocks.

[1] Department of Astronomy, University of Maryland, College Park, MD, USA. [2] Planetary Magnetospheres Laboratory, NASA Goddard Space Flight Center, Greenbelt, MD, USA. [3] Center for Space Sciences and Technology, University of Maryland, Baltimore County, Baltimore, MD, USA. [4] Center for Research and Exploration in Space Science and Technology II, NASA/GSFC, Greenbelt, MD, USA. ✉email: norberto.romanelli@nasa.gov

The solar wind (SW) can ideally be thought of as a plasma that is Maxwellian. This state can be disturbed by low frequency waves, in association with additional plasma populations. Low frequency waves are ubiquitous in space plasmas and have been observed in drastically different environments throughout the solar system. These waves can be the cause or effect of non-Maxwellian plasma velocity distribution functions observed in the pristine SW, as well as in solar system planetary magnetosheaths, ionospheres and foreshocks [e.g.,[1]]. For this reason, low frequency waves play a fundamental role in the transfer of energy and linear momentum between charged particles, especially in collissionless environments such as the SW or near a planetary body in a region known as the foreshock.

A planetary foreshock is the spatial region upstream of, but magnetically connected to, a planet's bow shock. As a result of this magnetic connection, the foreshock is filled with SW protons coexisting with a secondary population of backstreaming ions flowing upstream. The latter are produced by reflection of SW protons at the bow shock or leakage of plasma from downstream of the shock in the magnetosheath [e.g.,[2,3]]. Due to the presence of both ion populations, the local proton velocity distribution function is therefore non-Maxwellian. As the backstreaming particles move along the Interplanetary Magnetic Field (IMF) away from the planet, they constitute a source of free energy for various plasma instabilities. Such plasma instabilities are responsible for the presence of electromagnetic waves with magnetic field spectral power above the turbulent solar wind spectrum. These waves and the turbulent energy cascade rate contribute to restore the thermodynamical equilibrium of the plasma [e.g.,[4–9]].

Ultra-low frequency electromagnetic plasma (ULF) waves associated with backstreaming protons have been reported in several solar system planetary foreshocks [e.g.,[6,10–18]]. As shown by laboratory experiments, the Alfvénic Mach number (ratio between the solar wind velocity and the Alfvén speed) can have a strong impact on the presence of foreshock ULF waves. In particular, it was shown that the reflected ion current (proxy for a controlled backstreaming ion current) increases linearly with it, after a critical value is surpassed [e.g.,[19,20]]. In general, ion reflection is a fundamental property of relatively high Mach number collisionless shocks[3,21–24], although it is not the only variable affecting it [e.g.,[25]]. Mercury constitutes an ideal natural laboratory to test this theory and previous reports, as the solar wind around Mercury is characterized by relatively low SW Alfvénic Mach number (~3–6) and beta that vary with the planet's heliocentric distance [e.g.,[26–29]]. However, the SW Alfvénic Mach number at Mercury's heliocentric distance range is expected to be above the critical value (~2–3), thus particle reflection at the bow shock is not negligible[16,23,30]. Mercury's plasma environment properties are unique in the solar system, as the solar wind Alfvén Mach number is small but sufficient for the development of a foreshock. Moreover, the large range of heliocentric distances in its orbit allows us to study the wave occurrence variability in a way that is not possible elsewhere in the solar system unless we utilize a comparative planetology approach. Despite these factors, ULF waves around Mercury have only recently been studied statistically[30].

Previous studies have shown that ULF foreshock waves at Mercury can be classified in lower-frequency (~0.1–0.3 Hz), intermediate-frequency (~0.8 Hz) and higher-frequency (~2 Hz) waves[15,16,30]. The lowest and highest frequency waves display similar properties to the 30 s and 1 Hz waves detected at the terrestrial foreshock, respectively[17,31]. However, in contrast with the terrestrial foreshock,[16] found that the (lowest frequency) fast magnetosonic waves occur only sporadically at the Hermean foreshock, based on the analysis of a survey of ULF waves observed by MErcury Surface, Space ENvironment, GEochemistry, and Ranging (MESSENGER) Magnetometer (MAG) during a single Hermean foreshock passage[32,33]. In addition, the authors found that the most common wave phenomenon observed in the Hermean foreshock corresponds to higher ULF whistler waves.

[30] presented the first statistical analysis of the main properties of ULF waves associated with backstreaming ions observed in the Hermean foreshock, based on MESSENGER MAG observations. In particular, the authors found that waves with frequencies in the 0.05–0.41 Hz range in MESSENGER's reference frame, are close to being circularly polarized (although some elliptically polarized cases were also detected) and propagate quasi-parallel to the IMF (~10°). The authors also reported that these waves are characterized by a relatively small normalized wave amplitude when compared to their terrestrial counterpart and that the wave occurrence rate is relatively low, results likely associated with low backstreaming proton flux and/or variable external conditions. Moreover[30], also estimated that the backstreaming protons speed ranges between 0.95 and 2.6 times the SW speed, based on a resonance condition. This range is comparable to what has been observed at several solar system planetary foreshocks, thus suggesting that similar acceleration processes are occurring upstream of planetary bow shocks throughout the heliosphere[11]. An open question remains, however, in regards to understanding favorable conditions for generating these waves in planetary foreshocks.

In this work, we investigate the spatial region where 0.05–0.41 Hz ULF waves are most commonly detected in the foreshock of Mercury to determine which conditions are most favorable for the generation of these waves, resulting in the highest occurrence rate. By considering these results as a function of Mercury's heliocentric distance, we are able to apply them to the broader context of planetary foreshocks. We conclude that these quasi-monochromatic ULF waves occur preferably under relatively low strength and quasi-radial IMF. As a result, we find that the wave occurrence rate increases with Mercury's heliocentric distance. This trend is consistent with larger reflection of SW protons, associated with higher SW Alfvén Mach numbers.

## Results

**ULF Wave detection.** In this study we have analyzed all MESSENGER MAG measurements acquired upstream from Mercury's bow shock by utilizing the maximum sampling rate of 20 vectors/s[33] (see Methods, subsection The MESSENGER mission, for more details). Detection of backstreaming ion-generated ULF waves in the Hermean foreshock is based on both frequency and polarization properties[30]. We compute the Power Spectral Density (PSD) of the perpendicular and parallel components to the mean magnetic field [PSD($B_\perp$) and PSD ($B_\parallel$)] utilizing a Fast Fourier Transform algorithm. We perform this analysis over 204.8 s intervals with upstream MAG measurements. To consider the wave polarization properties, we also apply Minimum Variance Analysis (MVA) on MAG data[34].

Observations are analyzed in the solar foreshock coordinate system[35]. The foreshock coordinates used in this paper are DIST, $\theta_{BN}$ and $Z_0$. DIST measures the distance between MESSENGER and the bow shock location along the IMF direction. $\theta_{BN}$ is the angle between the IMF line that passes through the spacecraft location and the normal to the bow shock at the intersection point and is known to affect particle reflection [e.g.,[17,18]]. $Z_0$ measures the distance of MESSENGER perpendicular to the solar wind velocity- IMF ($V_{SW} - B_{IMF}$) plane that contains the center of Mercury. Figure 1 shows a schematic representation of the foreshock, where these variables are also presented. Hereafter, computed foreshock coordinates are the mean values over 204.8 s, over which PSD are also determined.

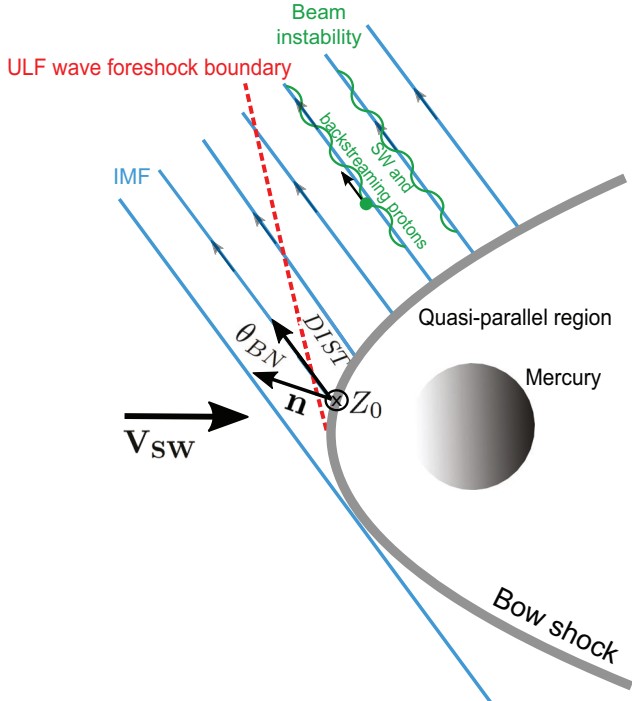

**Fig. 1 Illustration of the foreshock.** Schematic structure of the planetary foreshock for a given Interplanetary Magnetic Field (IMF) configuration. The solar wind (SW) flows from left to right with velocity $\mathbf{V_{SW}}$ and the bow shock is represented by the curved line. The bow shock structure depends on the angle between the shock normal ($\mathbf{n}$) and the magnetic field, $\theta_{BN}$. The quasi-parallel shock is typically defined as the region where $\theta_{BN} < 45°$. The foreshock, lying upstream of the bow shock and in the region downstream of the IMF tangent field line (first line from left to right), exhibits a complex structure where several regions and boundaries can be identified. The Ultra-low frequency (ULF) wave foreshock boundary displays the upstream limit of the ULF wave field inside the foreshock. Such wave field is partly created by plasma instabilities (such as the beam instability) and coexists with backstreaming protons (green dot) interacting with the incoming SW. The foreshock coordinates used on this paper are also shown for reference. *DIST* measures the distance between the spacecraft and the bow shock location along the IMF direction. $Z_0$ measures the distance of the spacecraft perpendicular to the SW velocity- IMF plane which contains the center of Mercury.

Based on the previous frequency, polarization, and connection analysis, our criteria state that a wave event has been detected if the spacecraft is connected to Mercury's bow shock and a peak in the PSD of the magnetic field measurements fulfills

$$max\left(PSD(\mathbf{B}_\perp)|_{\Delta f_2}\right) > r \; max(PSD(\mathbf{B}_\perp)|_{\Delta f_1}), \tag{1}$$

$$max\left(PSD(\mathbf{B}_\perp)|_{\Delta f_2}\right) > r \; max(PSD(\mathbf{B}_\perp)|_{\Delta f_3}), \tag{2}$$

and

$$Q_{75}(\lambda_2/\lambda_3) > \lambda_{23}, \tag{3}$$

where $\Delta f_1$, $\Delta f_2$ and $\Delta f_3$ are equal to $[0.0293-0.0488]$ Hz, $[0.0537-0.4150]$ Hz, $[0.4199-0.5957]$ Hz, respectively, and $Q_{75}(\lambda_2/\lambda_3)$ is the 75th quartile associated with the intermediate to minimum MVA eigenvalue ratio ($\lambda_2/\lambda_3$) distribution for a given 204.8 s interval. Thus, the value of $r$ is associated with the normalized power of the peak at the frequency range of interest, while $\lambda_{23}$ constitutes a threshold for an adequate application of the MVA. Combined, both parameters define the criteria for the detection of backstreaming ion generated ULF waves. The wave

selection criteria are described in more detail in Methods, subsections Power Spectral Density and Wave Polarization and Bow shock connectivity. An example of a positive wave detection can be found in Fig. 1 in[30], where $r = 4$ and $\lambda_{23} = 5$.

**Spatial Distribution**. We find that most of the ULF waves are observed for $DIST = [1.0 - 3.3]R_M$, $\theta_{BN} = [21.8-55.7]°$, and $Z_0 = [-3.4 \text{ to } 3.4]R_M$, despite MESSENGER's capability to detect them outside this spatial range. These results are shown in Fig. 2a–c that display the mean values of *DIST*, $\theta_{BN}$ and $Z_0$ as a function of Mercury's heliocentric distance for all 204.8 s intervals where MESSENGER was in the Hermean foreshock (in gray) and with ULF wave detection (in pink) meeting the criteria of $r = 2$ and $\lambda_{23} = 0$. It is also worth noticing that waves appear to be present up to $\sim5R_M$ both in *DIST* and $Z_0$, suggesting the presence of a limit from which waves are not longer present or detectable above the turbulent solar wind background. We find that such limit does not vary significantly with the explored wave selection criteria. In this study, we consider $r = 2$ and $\lambda_{23} = 0$ the minimum set of conditions to detect waves, avoiding detection of false positives. Interestingly, these quasi-monochromatic (clear dominant frequency) ULF waves are not detected above $\theta_{BN} \sim 70°$, suggesting that backstreaming ion fluxes fall off substantially with large $\theta_{BN}$. Moreover, it is worth mentioning the presence of a minimum solar wind convection time needed for wave growth, once these fluxes are significant. This result is consistent with previous reports related to other solar system planetary foreshocks [e.g.,[3,36,37]].

Given the differences in MESSENGER's foreshock sampling as a function of Mercury's heliocentric distances (gray dots and black bars, Fig. 2), hereafter we restrict our analysis to time intervals where MESSENGER was in the spatial range where most of the waves are detected (red horizontal dashed lines, Fig. 2), regardless of the presence of waves. This allows us to determine the variability in the wave occurrence rate along Mercury's orbit around the Sun, taking into account such spatial biases.

**Conditions Favorable for ULF Waves**. We find that the wave occurrence rate increases as a function of heliocentric distance (Fig. 3). This trend, observed throughout the entire MESSENGER mission, is present regardless of the explored wave selection criteria. We perform a linear fit for each of these curves, shown by the corresponding solid lines. The selection criteria and parameters associated with the best linear fits displayed in Fig. 3b are listed in Table 1. The linear correlation factors (R) and p-values suggest the correlation between the abundance of ULF waves and Mercury's heliocentric distance is significant.

Having taken into account spatial biases due to MESSENGER's orbit around Mercury, next we investigate the reason for the observed trends. In particular, we determine the variability of the wave's occurrence with the IMF intensity and IMF cone angle. The latter is defined as the angle between the IMF and the SW velocity, assumed to be radial. While the IMF intensity directly affects the Alfvén Mach number, known to strongly influence the backstreaming ion fluxes (likely source of these waves), the IMF cone angle influences the plasma instability with the largest linear wave growth rate and the foreshock size.

We find a decreasing trend with more waves detected for lower values of $B = |\mathbf{B_{IMF}}|$, for $B$ ranging between 10 nT and 50 nT, as shown in Fig. 4b. Because of its eccentric orbit and proximity to the Sun, the IMF at Mercury has a large spread of $B$ values and goes up much higher than any other planet in the solar system. By performing a polynomial fit we find that the wave occurrence rate $O(B)$ decays as $B^{-1.85}$, with a 95% confidence interval for the power law index equal to $[-2.14, -1.57]$. A decreasing trend with

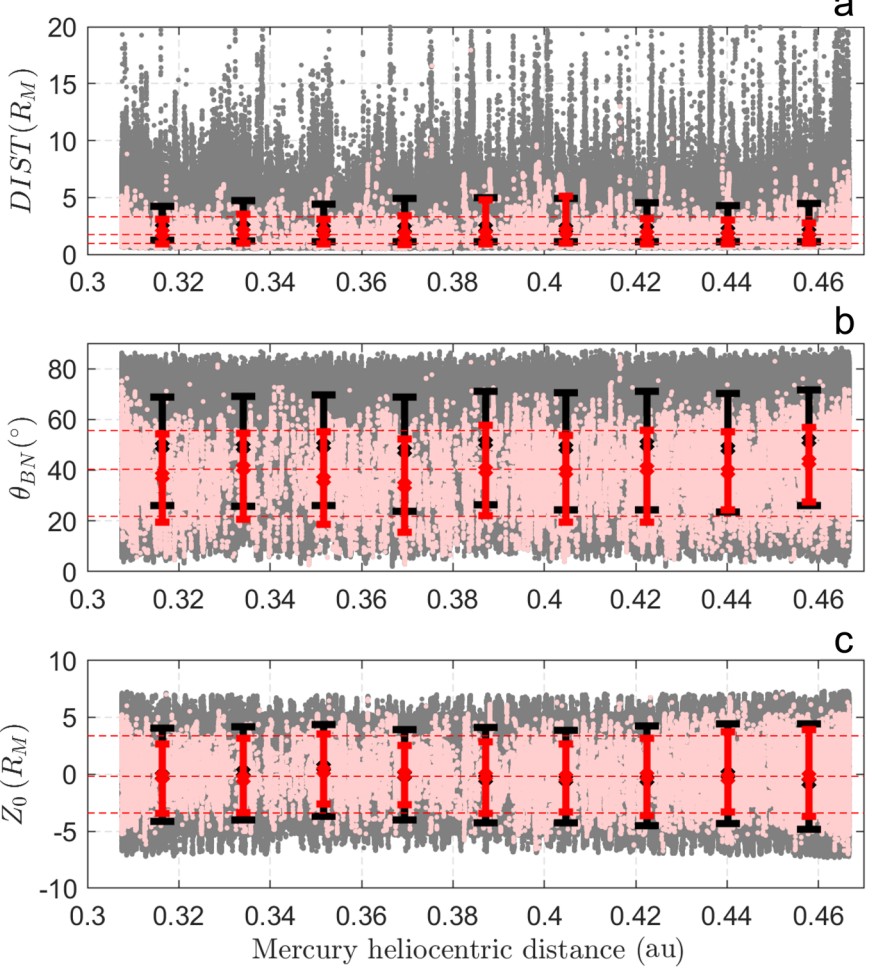

**Fig. 2 Spacecraft location.** Spatial coordinates of intervals in the Hermean foreshock (in gray) and with wave detection (pink) as a function of Mercury's heliocentric distance for $r = 2$ and $\lambda_{23} = 0$, for all corresponding MESSENGER observations obtained during approximately 17 Mercury years. **a** *DIST* measures the distance between MESSENGER and the bow shock location along the Interplanetary Magnetic Field (IMF) direction; **b** $\theta_{BN}$ is the angle between the IMF line that passes through the spacecraft location and the normal to the bow shock at the intersection point; **c** $Z_0$ measures the distance of MESSENGER perpendicular to the solar wind velocity- IMF plane that contains the center of Mercury. *r* is a factor associated with the intensity of magnetic field power spectral density at the wave frequency range of interest, and $\lambda_{23}$ constitutes a threshold for the intermediate to minimum eigenvalue ratio associated with Minimum Variance Analysis. The reader is referred to Eqs. (1)–(3) for a strict definition. Black and red vertical bars span the 10th to the 90th percentiles of the gray and pink data sets, respectively, within each bin (~0.0177 AU width). The middle point in each bar corresponds to the associated median. Horizontal red dashed lines are 10th, 50th, and 90th percentile of each spatial variable for all time intervals where waves are detected in the foreshock.

a similar power law index is obtained when the other selection criteria presented in Table 1 are applied. Figure 4d also shows that these ULF waves are seen more frequently for IMF cone angles smaller than ~35° and larger than ~140°. This preferential angular range and the peaks observed for a quasi-radial IMF are observed for all explored selection criteria.

Figure 5 shows the IMF intensity and cone angle as a function of Mercury's heliocentric distance for time intervals where MESSENGER is in the Hermean foreshock (inside the region where most of the waves were detected) and when waves are detected. Despite variability in the external conditions along Mercury's orbit (clearly shown by the black vertical bars), we find that the preference for low $B$ and IMF cone angles is observed throughout Mercury's heliocentric distance range. It is also worth noticing that while the background IMF intensity decreases with heliocentric distance as expected from Parker's model, the IMF cone angle displays a relatively much smaller statistical increase[38]. These authors reported very similar results for the background IMF intensity and cone angle around Mercury's perihelion and

aphelion. Therefore, although both parameters are found to affect the wave occurrence rate, the IMF intensity seems to be a key parameter behind the observed trend with Mercury's heliocentric distance. Indeed, a relatively low IMF intensity is present most of the time for relatively large heliocentric distances. In contrast, the IMF cone angle presents a slight increase with heliocentric distance but not as significant to overcome the effects that $B$ has on the wave occurrence rate.

As suggested by the results in Fig. 4, we report the presence of a clear anti-correlation between the wave occurrence rate and the IMF intensity throughout all MESSENGER mission, as can be seen from Fig. 6b, c. Such anti-correlation is the key factor responsible for the increase in the waves abundance around Mercury's aphelion. Although waves are seen most of the time for quasi-radial IMF (Fig. 4d), the range of this angular variable is not as clearly correlated with the planet's heliocentric distance, as shown in Figs. 5b and 6d. The best trigonometric fit in Fig. 6b is characterized by a period of ~88 days, an amplitude of 3.6% and a phase shift between Mercury's aphelion and the local maximum

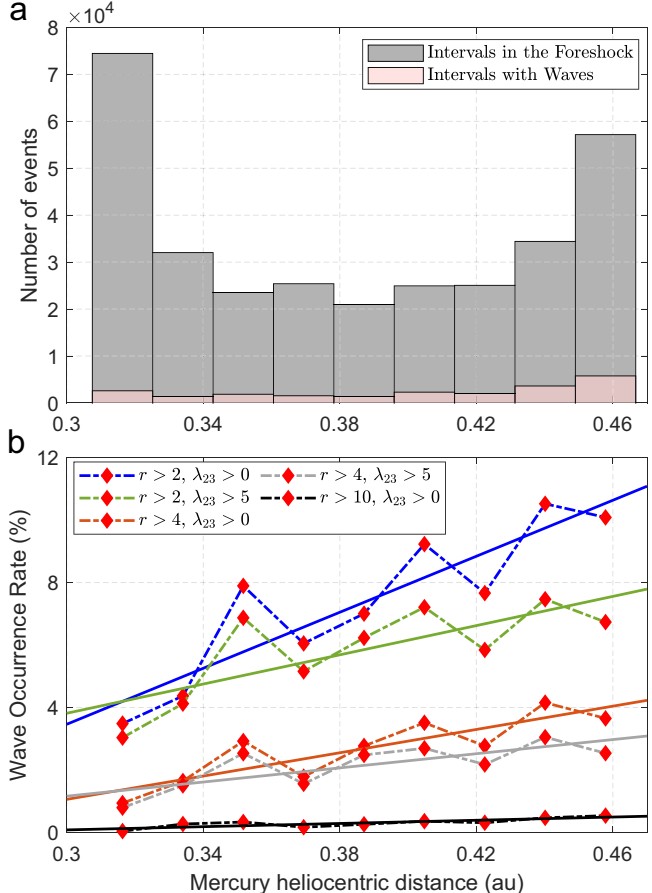

**Fig. 3 Wave Occurrence rate as a function of Mercury's heliocentric distance. a** Histograms of the number of selected intervals in the selected region inside the Hermean foreshock (gray) and with wave detection (pink) as a function of Mercury's heliocentric distance when $r = 2$, $\lambda_{23} = 0$; **b** Wave occurrence rate as a function of Mercury's heliocentric distance for five different selection criteria (dash curves) and their respective linear fits (solid line with the same corresponding color). $r$ is a factor associated with the intensity of magnetic field power spectral density at the wave frequency range of interest, and $\lambda_{23}$ constitutes a threshold for the intermediate to minimum eigenvalue ratio associated with Minimum Variance Analysis. The reader is referred to Eqs. (1)-(3) for a strict definition.

**Table 1 Linear fits. Parameters associated with the linear fits between the wave occurrence rate and Mercury's heliocentric distance, for different wave selection criteria. From left to right, $r$, $\lambda_{23}$, the linear correlation factor (R), the 95% confidence interval and the p-value associated with the linear fits shown in Fig. 3. $r$ is a factor associated with the intensity of magnetic field power spectral density at the wave frequency range of interest, and $\lambda_{23}$ constitutes a threshold for the intermediate to minimum eigenvalue ratio associated with Minimum Variance Analysis. The reader is referred to Eqs. (1)–(3) for a strict definition.**

| $r$ | $\lambda_{23}$ | R | 95% R-confidence interval | p-value |
|---|---|---|---|---|
| 2 | 0 | 0.89 | 0.57-0.98 | $1.1 \times 10^{-3}$ |
| 2 | 5 | 0.76 | 0.19-0.95 | $1.8 \times 10^{-2}$ |
| 4 | 0 | 0.86 | 0.46-0.97 | $2.8 \times 10^{-3}$ |
| 4 | 5 | 0.76 | 0.20-0.95 | $1.6 \times 10^{-2}$ |
| 10 | 0 | 0.84 | 0.40-0.97 | $4.6 \times 10^{-3}$ |

of the wave occurrence rate much smaller than the bin size (~0.4 h), supporting our previous conclusions. In addition, the Lomb-Scargle analysis shown in Fig. 7 displays a clear dominant peak with a period of ~88.2 days, which therefore differs by less than ~0.27% from Mercury's orbital period around the Sun. As can be seen in Fig. 6b, we also identify a significant decrease of the wave occurrence rate with respect to the mean of such fit (7.4%) around the end of the MESSENGER mission, possibly correlated with the general increase of $B$ associated with the maximum of the solar cycle 24.

Given that the ULF foreshock waves are present for a relatively small IMF cone angle range, it is possible to estimate the wave occurrence rate dependence on $B$ and Mercury's heliocentric distance ($r_{hc}$), independently. The employed methodology is the following. We first restrict our selected data set to cases where the IMF cone angle is between $[5.3-28.9]°$ and $[145.6-173.5]°$, that is to the 10th to 90th quartiles of this distribution. Next, we confirm that the wave occurrence rate increases with Mercury's heliocentric distance by performing linear fits analogous to the ones presented in Fig. 3. We find that linear correlation and p-values are larger than the ones reported in Table 1, supporting our previous results.

By assuming a polynomial dependence of the wave occurrence rate $O(r_{hc}, B)$ upon $B$ and $r_{hc}$, where the IMF cone angle range is fixed and temporal effects are neglected we propose

$$O = O(r_{hc}, B) \propto B^\alpha f(r_{hc}) \propto B^\alpha r_{hc}^\beta \qquad (4)$$

While Eq. (4) explicitly shows a dependence on $B$, $f(r_{hc})$ encompasses effects from hidden variables whose net effect we assume vary as $r_{hc}^\beta$. Therefore, Eq. (4) explicitly shows that $O$ is proportional to both $B$ and $r_{hc}$, with $\alpha$ and $\beta$ the corresponding polynomial index. Given that the IMF intensity decays with heliocentric distance as $B(r_{hc}) \sim r_{hc}^{-2}$, we also conclude that

$$O(r_{hc}, B(r_{hc})) \propto r_{hc}^\gamma \qquad (5)$$

with $-2\alpha + \beta = \gamma$. In other words, Eq. (5) shows that because of the decay of $B$ with $r_{hc}$, under these assumptions the wave occurrence rate is proportional to $r_{hc}$ to the power of $\gamma$. Moreover, the observed relationship between $O$ and $r_{hc}$, and the associated R and p-values associated with the linear fits analogous to the ones shown in Table 1 suggest that $\gamma \sim 1$. Such linear fits are the polynomial with the smallest number of degrees of freedom that fit the observations reasonably well. However, it is important to emphasize that $\gamma \sim 1$ can be affected by the explored heliocentric distance range and the presence of hidden variables and/or temporal variability affecting the values of wave's abundance. Despite these potential effects, we find that the wave's occurrence rate decreases with $B$ with a power law index ranging between $-2.35$ and $-1.80$, for different fixed ranges of $r_{hc}$. This result, in addition to the observed approximate linear increasing trend with $r_{hc}$, suggest that $\beta < 0$. Interestingly, a decreasing trend between the wave occurrence rate with $r_{hc}$, for a fixed value of $B$ and IMF cone angle suggests, in turn, the presence of another variable affecting the wave abundance. A proper determination of $\alpha$ and, in particular, of $\beta$ demands additional spacecraft observations and comparison with observations at other planetary foreshock when possible.

Figure 8 presents a scheme where the wave occurrence rate increases linearly with Mercury's heliocentric distance. At the same time, the background IMF intensity decreases, affecting the Alfvénic Mach number, a parameter known for affecting the backstreaming proton fluxes, the most likely source for the ULF waves studied in this work.

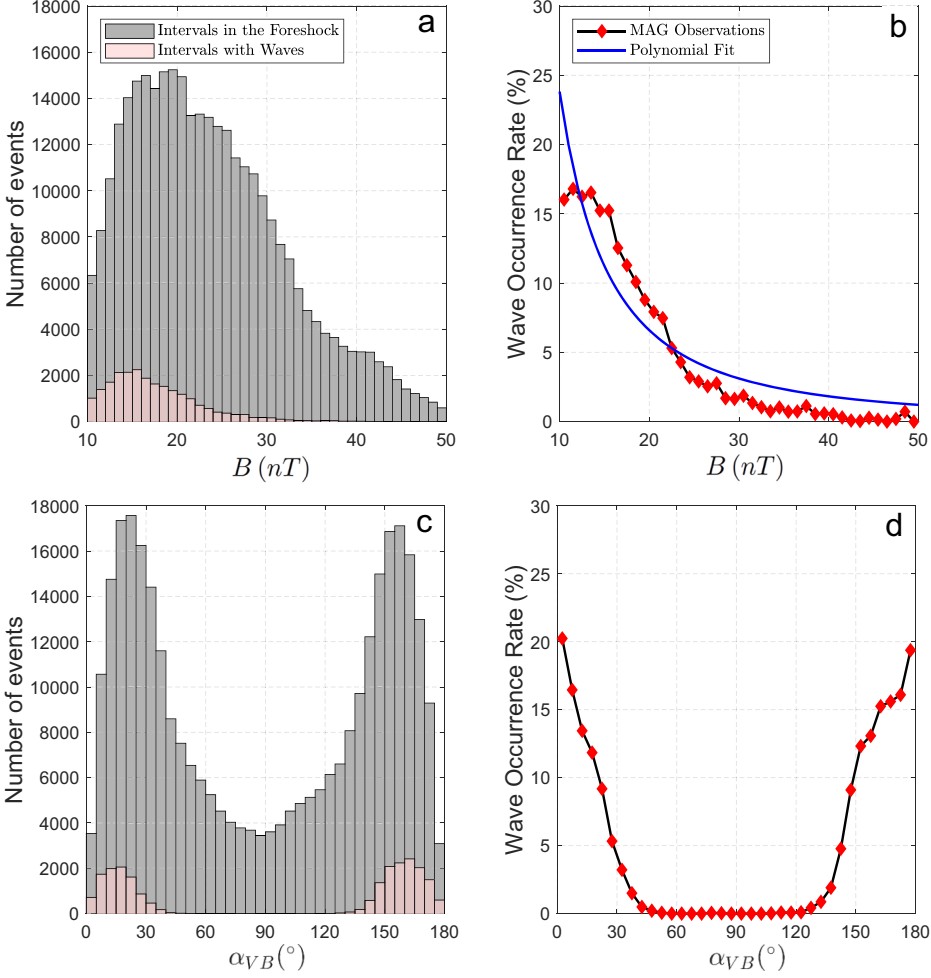

**Fig. 4 Wave occurrence rate as a function of Interplanetary Magnetic Field (IMF) strength and cone angle. a**, **c** Histograms of the number of selected intervals in the Hermean foreshock (gray) and with wave detection (pink) as a function of the (**a**) IMF intensity and (**c**) IMF cone angle when $r = 2$, $\lambda_{23} = 0$; **b**, **d** Associated wave occurrence rate as a function of the (**b**) IMF intensity (in black) and as a function of the (**d**) IMF cone angle. Best polynomial fit associated with wave occurrence rate as a function of IMF intensity is shown in blue (**b**). $r$ is a factor associated with the intensity of magnetic field power spectral density at the wave frequency range of interest, and $\lambda_{23}$ constitutes a threshold for the intermediate to minimum eigenvalue ratio associated with Minimum Variance Analysis. The reader is referred to Eqs. (1)–(3) for a strict definition.

## Discussion

In this work we have analyzed the dependence of the 0.05–0.41 Hz ULF wave occurrence rate on the IMF intensity and cone angle, taking into account spatial biases associated with MES-SENGER trajectory. Both parameters are relevant since they affect the foreshock size, the linear wave growth rate of the plasma instabilities and the Alfvén Mach number. The latter parameter can affect the solar wind backstreaming proton fluxes responsible for the analyzed ULF waves. Waves occur preferably for quasi-radial IMF configuration, increasing the foreshock size where waves can grow. The most likely instability responsible for these waves is the ion-ion right hand resonant [e.g.,[39,40]]. Indeed, wave polarization and frequency range are consistent with such instability and are similar to 30-s waves previously reported at the terrestrial foreshock[30]. Waves also occur preferably for relatively low IMF intensity values, that is, for relatively high Alfvén Mach numbers. Higher Alfvén Mach numbers are generally associated with increasing particle reflection at the bow shock[3,21–24]. In particular, Fig. 6b, c display a significant decrease in the wave occurrence rate associated with the relatively large $B$ values, that took place around the maximum of the solar cycle 24.

Interestingly, detection of the ULF waves throughout Mercury's orbit suggests that the conditions for backstreaming ions are potentially present for all of Mercury's heliocentric distances.

Once excited, large-scale magnetosonic ULF waves are convected by the solar wind and transmitted through the bow shock. These waves have been detected in the downstream region of several solar system planetary magnetospheres and constitute an important source of wave energy [e.g.,[41,42]]. They are capable of propagating into the magnetosphere, heat the ionosphere, launch bursts of time-dispersed energetic ions and also affect escape of heavy ions from the planetary atmospheres, among other outcomes [e.g.,[43–45]]. Thus, we can expect significant energy transfer into the Hermean magnetosphere by ULF magnetosonic waves, despite the intrinsic properties of this planet (e.g., the lack of an ionosphere and the relatively large conductive core) and the highly variable external conditions [e.g.,[46]]. It is also worth mentioning that whistler waves seem to occur more frequently in the Hermean foreshock[16]. The determination of the physical mechanisms associated with such coupling and the relative energy contribution compared to other frequent processes in the Hermean magnetosphere, such

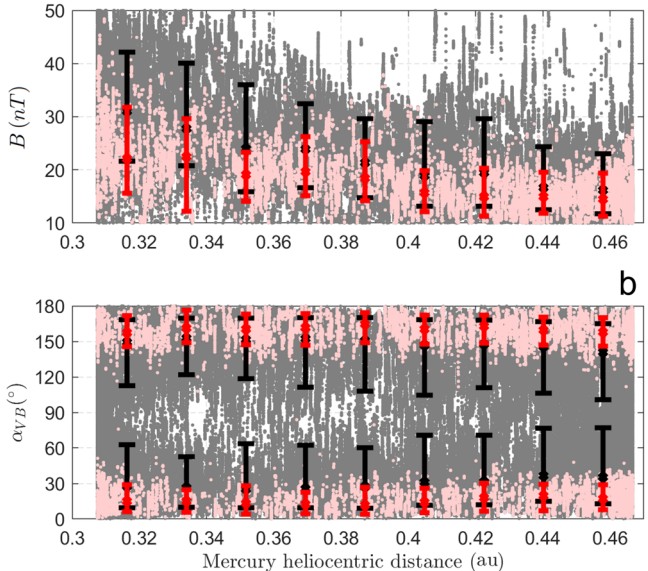

**Fig. 5 External conditions.** Interplanetary Magnetic Field (**a**) intensity and (**b**) cone angle for all intervals in the Hermean foreshock (in gray) and with wave detection (pink) as a function of Mercury's heliocentric distance for $r = 2$ and $\lambda_{23} = 0$, for all corresponding MESSENGER observations obtained during approximately 17 Mercury years. $r$ is a factor associated with the intensity of magnetic field power spectral density at the wave frequency range of interest, and $\lambda_{23}$ constitutes a threshold for the intermediate to minimum eigenvalue ratio associated with Minimum Variance Analysis. The reader is referred to Eq. (1)–(3) for a strict definition. Black and red vertical bars span the 10th to the 90th percentiles of the gray and pink data sets, respectively, within each bin (~0.0177 AU width). The middle point in each bar corresponds to the associated median.

as, magnetic reconnection[47], define an interesting topic for future studies.

We also report the presence of a long term trend ( ~ 1 Mercury year) where the abundance of foreshock ULF waves, likely generated by backstreaming protons, varies with Mercury's heliocentric distance. Our results suggest that such variability is partly associated with the decay of the IMF intensity radially from the Sun. Moreover, the characterization performed by means of Eqs. (4) and (5) suggests the presence of an additional variable affecting the waves abundance. One possible candidate is the solar wind proton density. Indeed, such density decreases with heliocentric distance, thus decreasing the Alfvén Mach number for a relatively constant solar wind speed. Therefore, a linear dependence of the wave occurrence rate with $r_{hc}$ (as shown in Fig. 3), could be associated with a linear dependence on the Alfvénic Mach number, at least for the explored heliocentric distance range. However, it is important to emphasize that the solar wind proton density does not only affect the Alfvén Mach number. Indeed, it can also affect the linear wave growth rate and the flux of reflected solar wind protons at the Hermean bow shock. Thus, deviations from a linear dependence on the Alfvén Mach number could be present despite the results displayed in Fig. 3.

Differences in the linear fits for different wave selection criteria shown in Fig. 3b could be related to the temporal evolution of the waves. Indeed, even though the most likely source of all observed fast magnetosonic waves are backstreaming ions, the waves that present a clear peak in the PSD and a well-defined polarization state (e.g., $r = 2$, $\lambda_{23} = 5$) are likely in an earlier evolution stage than those that only kept the peak in the PSD (e.g., $r = 2$, $\lambda_{23} = 0$)

[see, e.g.,[48,49], and references therein]. If this is the case, then the derived increase in the slope of the wave occurrence rate as a function of Mercury's heliocentric distance (for a fixed $r$ value, but for decreasing values of $\lambda_{23}$), can be understood in terms of the time needed to reach a given wave stage. In other words, wave events satisfying $r = 2$ and $\lambda_{23} = 5$ are likely in an early stage, while wave events satisfying $r = 2$ and $\lambda_{23} = 0$ (being a set of wave events that contains the previous one) is associated with both, early stage and more evolved waves. Given that the development of the latter most likely require more time with the appropriate external conditions, the fits shown in Fig. 3b suggest that the conditions favorable for these ULF waves occur more frequently as Mercury's heliocentric distance increases. In addition, we find the wave occurrence rate decreases for increasing values of $r$. This can be understood in terms of the necessary energy input and the time needed to reach relatively high amplitude waves. Indeed, Fig. 3b shows a gradual decrease in the wave occurrence rate as a function of Mercury's heliocentric distance for $r$ ranging between 2 and 10. This could be indicative of a limit to the maximum amplitude achievable in the Hermean foreshock. Bepi–Colombo magnetic field and particle measurements in the foreshock of Mercury are of utmost importance in order to test these conclusions and to better determine the dependence of the wave occurrence rate with the solar wind proton density and other parameters[50].

Finally, although the conditions (low Alfvén Mach number, low plasma beta) around Mercury are not seen very frequently around other planetary foreshocks in the solar system, this analysis allows us to identify the dependence of the wave occurrence rate on key parameters affecting the SW proton reflected fluxes in a planetary bow shock. The relationship between observed wave frequency and IMF intensity reported in[11] and[30] suggests that there are indeed similar acceleration mechanisms taking place in all solar system planetary foreshocks. The upstream region of interplanetary shocks constitute another plasma environment where upstream waves are observed under low Mach number regimes[51]. In particular, ULF waves similar to the ones analyzed in the present study have been reported by[52] and[53]. Interestingly, the absence of steepened waveforms and shocklets upstream from low Mach number interplanetary shocks[54,55] is compatible with previous analysis on the Hermean foreshock[16] and the relatively low normalized wave amplitude and direction of wave propagation found in[30]. Moreover, the conditions around Mercury are expected to be similar, in some respects, to the ones of several close-in exoplanets around red dwarfs, the most common type of star in the Milky Way. This is the case, for example, for several exoplanets in the TRAPPIST 1 system under conditions of minimum stellar wind total pressure[56]. Although density and solar wind velocity regimes are clearly different, the Alfvénic Mach number is in the same order. If this is the key parameter affecting the backstreaming ions we can expect a similar process taking place around those exoplanets.

## Methods

**The MESSENGER mission.** MESSENGER was inserted into orbit around Mercury on 18 March 2011. The spacecraft's orbit had high eccentricity, a ~12-h period, and 82° inclination[32]. The apoapsis altitude and orbital period were reduced on 16 April 2012 to ~4.1 $R_M$ and ~8 h, respectively ($R_M$ stands for Mercury's radii equal to 2440 km). However, as shown in Fig. 6a, the spacecraft provided magnetic field measurements in the Hermean foreshock throughout all the mission. The reader is referred to[57,58] for additional information on MESSENGER's trajectory relative to Mercury's magnetopause and bow shock location.

**Bow shock connectivity.** The methodology employed for the detection of foreshock ULF waves is similar to the one considered in[30]. The determination of the connection of MESSENGER to the Hermean bow shock (for each 204.8 s interval) is done based on the solar foreshock coordinates introduced by[35], and the average

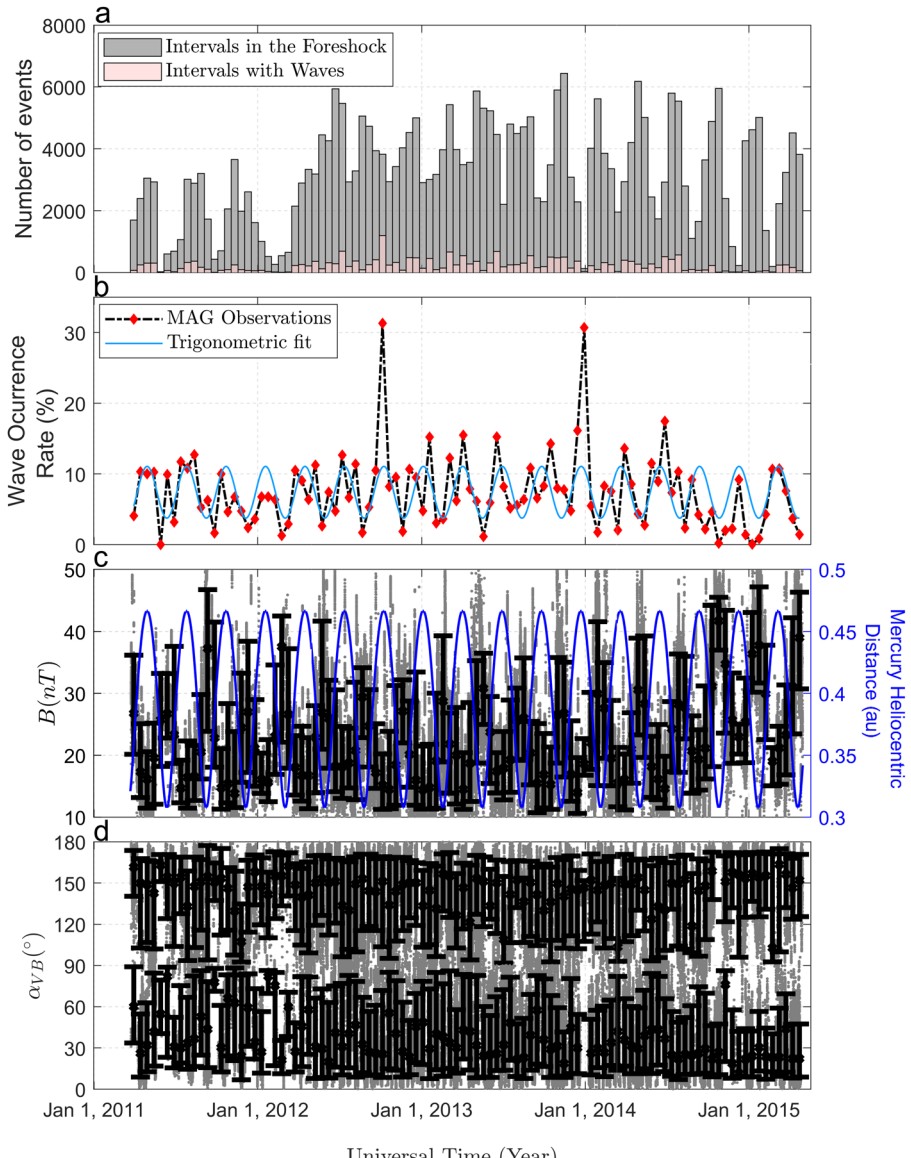

**Fig. 6 Wave occurrence as a function of time. a** Histograms of the number of selected intervals in the Hermean foreshock (gray) and with wave detection (pink) as a function of time when $r = 2$, $\lambda_{23} = 0$; **b** Associated wave occurrence rate as a function of time based on Magnetometer observations, and best trigonometric fit (light blue); **c**, **d** Interplanetary Magnetic Field intensity and cone angle as a function of time, for all intervals in the Hermean foreshock (in gray), black vertical bars correspond to the 10th, 50th, 90th percentiles for 15-day bins; (**c**, right vertical axis) Mercury's heliocentric distance as a function of time (in blue). $r$ is a factor associated with the intensity of magnetic field power spectral density at the wave frequency range of interest, and $\lambda_{23}$ constitutes a threshold for the intermediate to minimum eigenvalue ratio associated with Minimum Variance Analysis. The reader is referred to Eqs. (1)-(3) for a strict definition.

bow shock fit computed in[57]. For this, we perform a transformation from the aberrated Mercury solar magnetospheric (MSM) coordinates[30,59] to the latter system based on[35]. To determine the foreshock coordinates and bow shock connectivity associated with each interval, we compute the IMF vector and MES-SENGER location in the aberrated MSM coordinate system[30]. We conclude that MESSENGER was connected to the Hermean bow shock during each 204.8 s interval, if the IMF line at its mean position intersects the bow shock average fit. Since the location of this boundary changes in response to variability in the solar wind conditions, we increase the reported estimate of the semi-latus rectum up to 30%. Taking this consideration into account, the employed methodology allows us to analyze events that took place upstream from the Hermean bow shock.

**Power spectral density and wave polarization**. Each of the analyzed 204.8 s time windows contains a minimum of ~10 wave periods (wave frequency varies with IMF intensity) and comprises 4096 magnetic field observations. The length of these intervals allows us to compute PSD($\mathbf{B}_\perp$) and PSD ($\mathbf{B}_\parallel$) by means of a Fast Fourier Transform algorithm. Given that we utilize the maximum sampling rate of MAG

(20 Hz), the associated spectral resolution $\Delta f$ is 0.00488 Hz. We consider a sliding window, such that the intersection between neighboring time intervals is 87.5%. Definition of $\Delta f_1$, $\Delta f_2$ and $\Delta f_3$ frequency intervals are based on previous reports [e.g.,[11,15,16,30] and numerical simulations[60]. In particular,[11] and[30] have shown that the observed wave frequency of the foreshock fast magnetosonic waves under study displays a linear dependence with the IMF intensity for several solar system planetary magnetospheres. These observed wave frequencies may therefore indicate a dependence on local ion gyrofrequencies (see, for instance, Fig. 3b in[30]). Our analysis is limited to cases in which the strength of the mean IMF for a given time interval is equal to or greater than 10 nT, allowing us to detect ULF waves with observed frequencies in the $\Delta f_2$ range.

The wave polarization analysis is performed making use of the MVA technique[30]. By considering sub-intervals on the order of ~4 observed wave periods (~14 s to 50 s), we are able to determine the corresponding median values and percentiles of the eigenvalues ratios, for each of the 204.8 s time interval. As shown in Eq. (3), our wave selection criteria specifically considers the 75th quartile of the $(\lambda_2/\lambda_3)$ distribution.

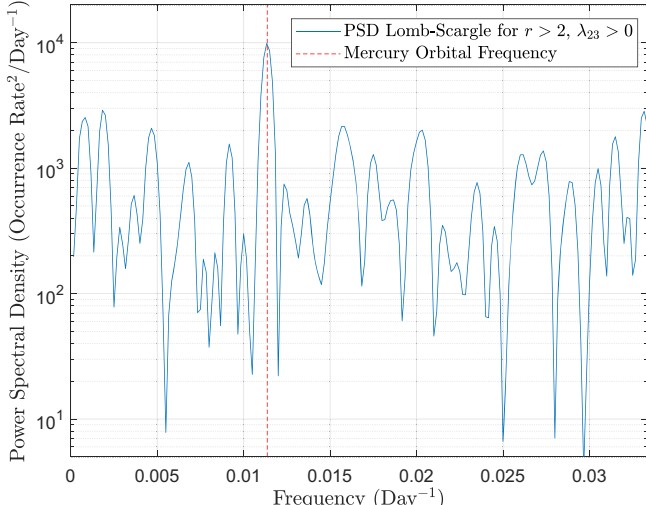

**Fig. 7 Lomb-Scargle analysis.** Power Spectral Density (Lomb-Scargle analysis) of the wave occurrence rate ($r > 2$, $\lambda_{23} > 0$, shown in Fig. 6b) as a function of frequency. Vertical red dashed line corresponds to Mercury's orbital frequency around the Sun. $r$ is a factor associated with the intensity of magnetic field power spectral density at the wave frequency range of interest, and $\lambda_{23}$ constitutes a threshold for the intermediate to minimum eigenvalue ratio associated with Minimum Variance Analysis. The reader is referred to Eqs. (1)–(3) for a strict definition.

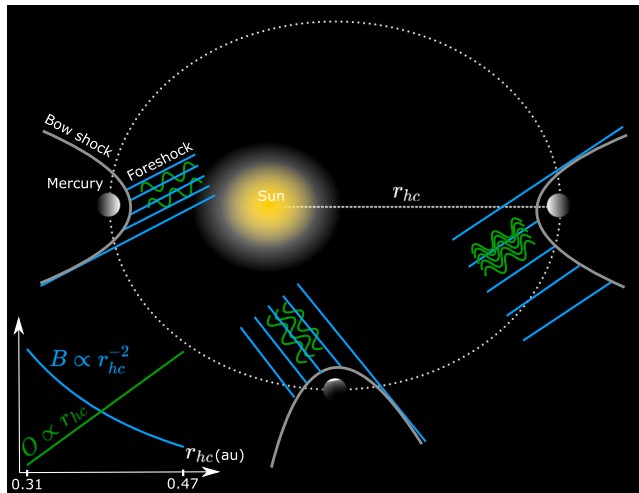

**Fig. 8 Illustration of the main result.** Schematic representation of the variability of the wave occurrence rate ($O$) and the Interplanetary Magnetic Field (IMF) intensity ($B$) as a function of Mercury's heliocentric distance ($r_{hc}$). The IMF intensity, approximately proportional to $r_{hc}^{-2}$, is represented by the distance between light blue lines upstream of each bow shock. In this work, we find that the $O$ increases approximately as $r_{hc}$, with Mercury's heliocentric distance (represented by the number of green curves upstream of each bow shock). These results are also summarized in the inset plot. These trends are consistent with larger reflection of solar wind protons for larger heliocentric distances, associated with higher solar wind Alfvén Mach numbers. The backstreaming protons are the most likely source for the analyzed waves.

The results presented in this work correspond to $r$ ranging between 2 and 10, and $\lambda_{23} = 0$ or $\lambda_{23} = 5$, as shown in Table 1. These criteria are justified by the following argument. The presence of other waves, such as whistlers at ~2 Hz, or ~0.8 Hz will not strongly affect the results derived from magnetic field power spectra given the clear shift in the observed frequencies. Therefore, the value of $r$ (PSD criteria) can be varied significantly. However, MVA analysis can be affected

by the presence of other wave modes, specially when the power in one or both of these modes is on the order or larger than the waves under study. It is because of this and the size of the chosen time intervals that the MVA analysis is applied up to $\lambda_{23} = 5$. A similar criteria has also been used to analyze ULF waves in region upstream from the Martian bow shock[49,61,62].

## Data availability

MESSENGER MAG data used in this study are publicly available through the Planetary Data System (https://pds-ppi.igpp.ucla.edu/search/view/?f=yes&id=pds://PPI/MESS-E_V_H_SW-MAG-3-CDR-CALIBRATED-V1.0/DATA/MSO). This data set can also be accessed following the path: /home/Mercury/Messenger/Magnetometer/MESSENGER MAG Calibrated Data Bundle/data/mso/. The magnetic field data generated in this study have been deposited in https://figshare.com/s/e0b7d259afb8b01cf308. The data that support the findings of this study are available from the corresponding author upon reasonable request. Source data are provided with this paper.

## Code availability

The code used in this analysis is based on four main routines. The Fast Fourier Transform (https://www.mathworks.com/help/matlab/ref/fft.html) and Lomb-Scargle periodogram (https://www.mathworks.com/help/signal/ref/plomb.html) developed by MATLAB, a Minimum Variance Analysis developed following[34] and bow shock connectivity routine developed following[35]. The code is available upon reasonable request.

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

## Acknowledgements

N.R. is supported through a cooperative agreement with Center for Research and Exploration in Space Sciences & Technology II (CRESST II) between NASA Goddard Space Flight Center and University of Maryland College Park under award number 80GSFC21M0002. Support for this research was also provided by GSFC/EIMM within NASA's Planetary Science Division Research Program. N.R. and G.A.D. were also supported by the NASA ROSES Solar System Workings program (NNH19ZDA001N-SSW), proposal # 19-SSW19-0106. N. R. would like to thank Maíra Dutra for the contributions and time she dedicated to create the schematics presented in Figs. 1 and 8.

## Author contributions

N.R. led the work and conducted most of the analysis of MESSENGER MAG dataset. N.R. and G.A.D. contributed to scientific interpretation of the results and drafting of the manuscript. All authors discussed the results and conclusions of the manuscript.

## Competing interests

The authors declare no competing interests.
