## [Peer Review File · Nature Communications]

REVIEWER COMMENTS

Reviewer #1 (Remarks to the Author):

The paper by Romanelli and DiBraccio uses MESSENGER magnetometer data to study the occurrence of ULF waves in Mercury's foreshock. Specifically they look at the effects which the orientation and magnitude of the interplanetary magnetic field has on the generation of such wave activity. It is concluded that there are preferential IMF cone angles, where more waves are detected when the field is close to being oriented radially towards or away from the Sun. There is also a more significant dependence upon the magnitude of the IMF, where more waves are detected when $|B|$ is lower.

Overall the article is well written and logically organized. The data are accessible via the provided link (although a little navigation is required). The methods of data analysis used by the authors are well-founded, being based upon methods used their previous work and the work of others. I would recommend this paper for publication, with a few comments and minor corrections to consider (see below).

14 (and occurrences elsewhere): Alfven -> Alfvén

45 (and occurrences elsewhere): Alfvenic -> Alfvénic

53 - 57: Please state the critical Mach value and also the typical Mach range for the solar wind in the vicinity of Mercury's orbit for a little context.

66: Please provide the references for MESSENGER and MAG acronyms

MESSENGER: Solomon, R. L., Sean, C., McNutt, R. E. G., & Domingue, D. L. (2007). MESSENGER mission overview. *Space Science Reviews*, 131(1), 3–39. <https://doi.org/10.1007/s11214-007-9247-6>

MAG: Anderson, B., M. Acuña, D. Lohr, J. Scheifele, A. Raval, H. Korth, and J. Slavin (2007), The Magnetometer instrument on MESSENGER, *Space Sci. Rev.*, 131(1–4), 417–450, doi:10.1007/s11214-007-9246-7.

64-73: I think it would be good to state what sort of frequency and polarizations that the waves of Romanelli et al. 2020 were characterized by somewhere here.

80-129: While reading this section initially I had some questions about the location of the bow shock which would be required for the B vector transform - I wanted to know whether a model was used (e.g. Slavin et al. 2009 or Winslow et al. 2013) or whether the position was obtained directly using the MAG data as MESSENGER traversed the bow shock. I was also interested in the rationale behind the length of the window used and the frequency bands. It wasn't until I had read the "Methods" section after reading the conclusion that those questions were answered. It would be good to either merge the information from both sections such that all of the information about the methodology is presented before the results, or to reference the "Methods" section from within the "Sampling Mercury's foreshock with MESSENGER MAG" section where appropriate.

105: Are the perpendicular components combined prior to the FFT analysis? If so, would you run the risk of losing information using this method? If there is a circularly polarized oscillation with a constant amplitude in the perpendicular components, then surely that would manifest in a constant B_{\perp} and not be picked up by the FFT?

107: Please define what the sub-interval is. (i.e. is it some fraction of the 204.8 s interval? If so, how long are they?)

136-138: is it possible that this limit appears as a result of the wave selection process?

Figure 6b - While I like the look of this plot, how confident are you in the fit? A tiny part of me worries that the nice blue line might just be guiding my eye to believe that I see an 88-day pattern in the wave occurrence rate, when there are obviously other periods present. A good thing to show here would be a FFT/Lomb-Scargle analysis of this plot to show that there is a peak at the orbital period of Mercury.

193-196: What is the phase shift/how many days out of phase?

314: It might be a good idea to link directly to the MAG data.

Reviewer #2 (Remarks to the Author):

The manuscript is well written and presents interesting results about ULF waves in the Hermean environment, namely the foreshock. The conclusions are relevant for the space plasmas community as they give insight and enhance current understanding of how wave power varies with plasma conditions in shock related regions. The paper can be accepted for publication after some points described below have been clarified.

Line, 7. While planetary shock properties depend on solar wind parameters, the sentence “to elucidate the transfer of energy from the Sun to planetary systems “is misleading on this line.

Line 41, ULF waves are not “a type” of electromagnetic plasma waves, several modes can be generated in foreshock regions.

Line 54, “low Mach number” how low?

Line 105, Bcomp refers to parallel?, specify, why not use “paral”, or the parallel sign

Line 141, authors claim that ULF waves are not observed for a perpendicular shock because they do not have time and space to grow, this is misleading, ion escape from the shock is dramatically reduced after a certain value of oblique θ_{Bn} , so instability conditions for grow do not exist. Some words about variation of ion fluxes with shock geometry need to be included here and relate to conditions of instability, etc

Line 244, needs rewording, as it is, it seems that waves increase the Mach number, not the value of B.

Line 268, as the authors state, planetary bow shocks in the solar system have larger Alfvén Mach number than at Mercury. However, interplanetary shocks can have low Mach numbers and nothing is mentioned about them. A few lines related to these shocks and results present in this study are needed.

Minor

Line 14 Alfvén Mach --- > Alfvén Mach number

Reviewer #3 (Remarks to the Author):

This paper investigates the generation of fast magnetosonic plasma waves observed over four years with the MESSENGER magnetometer in the foreshock region of Mercury. The waves are in the 0.05-0.41 Hz frequency range and are most likely produced by cyclotron resonance between the solar wind plasma and backstreaming ions reflected from the Hermean bow shock. The authors show that these waves occur preferentially <5 RM upstream under quasi-radial conditions, and with lower IMF intensity and larger heliocentric distance. There is a suggestion some other variable also influences wave occurrence.

The authors have analysed a very large data set and drawn inferences which provide new insight on the generation of low frequency plasma waves in planetary foreshocks. This is both topical and timely work for planetary scientists. The manuscript is generally well written and the figures are well presented. However, I have some concerns regarding the scope of the data interpretation, and minor comments about typos, etc. These are listed below. The paper should be accepted for publication once these have been addressed.

Main points

1. The paper focuses on magnetosonic foreshock waves in the 0.05-0.41 Hz frequency range.

(a) Please explain or justify this choice of frequency.

(b) Le et al., J. Geophys. Res. 118, 2013, found that these waves are relatively weak and occur only sporadically in Hermean foreshock (likely because of the low Alfvénic Mach number), while ~1 Hz whistler mode waves are much more abundant in the foreshock, sometimes appearing in wavetrains lasting hours. Figures 3(a), 4(a) and 6(a) in the present paper clearly show the occurrence of the magnetosonic waves is very low, but the manuscript is silent regarding the 1 Hz whistler waves. Please clarify the occurrence and significance of the two wave populations, especially in regard to the discussion and conclusions about waves in planetary foreshock regions.

2. Data selection and analysis is based on comparison of peak power spectral density in the frequency range of interest with that in lower and higher frequency bands. The ratio of these values is given by r . For the figures presented in this paper $r=2$, while in Romanelli et al (2020) $r=4$. Figure 3(b) shows that wave occurrence with heliocentric distance is quite different for $r=10$ (a stricter selection criterion) than for $r=2$. An additional selection criterion is the ratio of intermediate and minimum eigenvalues of the covariance matrix of the magnetic field in the given time interval, λ_2/λ_3 ($=5$ in Romanelli et al 2020). Please justify this choice of selection criteria. For example, would there be any change in the wave occurrence rate with IMF intensity, shown in Figure 4(b), for higher r or different λ values?

3. For Earth the magnetosonic waves generated in the foreshock have period around 30 sec and propagate into the magnetosheath and magnetosphere where they provide an importance source of wave energy. Please comment on what could be expected for Mercury, and indeed other planets with similar solar/stellar wind environments.

4. On page 14, lines 249, 250 is the statement “detection of the ULF waves throughout Mercury’s orbit suggests the presence of a relatively stable backstreaming ion population.” This is misleading for two reasons. First, the solar wind at Mercury is highly variable so would not produce a ‘relatively stable’ backstreaming population. Second, the Mach number at Mercury is fairly small so the backstreaming ion beams are weak (compared with Earth’s foreshock) and the resultant waves are much smaller than at Earth. Please amend the text and discussion accordingly. See e.g. Zurbuchen et al., Adv. Space Res., 33, 2004, and Le et al, 2013.

Other points

Page 2, line 48. “... with, after...” Word missing

Page 5, line 115. “... our criteria state that”

Page 6, line 123. “... 204.8 s interval”

Page 6, line 129. “... Romanelli et al (2020), where $r=4$ and $\lambda_{2,3}=5$.”

Page 6, line 147. “... This allows us”

Page 7. Figure 2 caption. “... The middle point in each bar corresponds...”

Page 7, lines 151, 152. The statement that the trend for wave occurrence to increase with heliocentric distance regardless of the wave selection criteria is questionable. The trend decreases with increasing r and is hardly present for $r=10$.

Page 9, line 189. “... through all the MESSENGER mission”

Page 13, line 225. "... for a fixed value"

Page 14, lines 238, 239. "Wave are seen most of the time...." This statement could be misleading. Waves do occur preferably under these conditions but wave occurrence is never high.

Page 14, line 243. "Waves are seen most of the time..." Same comment as above.

This is our answer to the referee's reports of the article entitled: 'Variability of Ultra-Low Frequency Waves Occurrence in Mercury's Foreshock'.

We thank the reviewers for devoting their time to review our paper, and for their encouraging comments and useful suggestions. Their comments have helped us to improve the clarity and state of the manuscript. In the following, they can find our detailed answers (in blue) inserted after each specific point. The line numbers indicated in the response to the reviewers are associated with the line numbers of the new version of the manuscript. This version also presents modifications in the structure of the manuscript, to adhere to the formatting requirements of Nature Communications.

Yours sincerely,

Norberto Romanelli and Gina A. DiBraccio.

REVIEWER COMMENTS

Reviewer #1 (Remarks to the Author):

The paper by Romanelli and DiBraccio uses MESSENGER magnetometer data to study the occurrence of ULF waves in Mercury's foreshock. Specifically they look at the effects which the orientation and magnitude of the interplanetary magnetic field has on the generation of such wave activity. It is concluded that there are preferential IMF cone angles, where more waves are detected when the field is close to being oriented radially towards or away from the Sun. There is also a more significant dependence upon the magnitude of the IMF, where more waves are detected when $|B|$ is lower.

Overall the article is well written and logically organized. The data are accessible via the provided link (although a little navigation is required). The methods of data analysis used by the authors are well-founded, being based upon methods used their previous work and the work of others. I would recommend this paper for publication, with a few comments and minor corrections to consider (see below).

14 (and occurrences elsewhere): Alfven -> Alfvén **Corrected.**

45 (and occurrences elsewhere): Alfvenic -> Alfvénic **Corrected.**

53 - 57: Please state the critical Mach value and also the typical Mach range for the solar wind in the vicinity of Mercury's orbit for a little context. **Thank you, we have included this information in the new version of the manuscript.**

66: Please provide the references for MESSENGER and MAG acronyms

MESSENGER: Solomon, R. L., Sean, C., McNutt, R. E. G., & Domingue, D. L. (2007). MESSENGER mission overview. *Space Science Reviews*, 131(1), 3–39. <https://doi.org/10.1007/s11214-007-9247-6>

MAG: Anderson, B., M. Acuña, D. Lohr, J. Scheifele, A. Raval, H. Korth, and J. Slavin (2007), The Magnetometer instrument on MESSENGER, *Space Sci. Rev.*, 131(1–4), 417–450, doi:10.1007/s11214-007-9246-7.

Thank you, we have provided references to these papers in this paragraph, in addition to the ones that were included in the next sections of the paper.

64-73: I think it would be good to state what sort of frequency and polarizations that the waves of Romanelli et al. 2020 were characterized by somewhere here. Thank you, we have included this information in Lines 71-76.

80-129: While reading this section initially I had some questions about the location of the bow shock which would be required for the B vector transform - I wanted to know whether a model was used (e.g. Slavin et al. 2009 or Winslow et al. 2013) or whether the position was obtained directly using the MAG data as MESSENGER traversed the bow shock. I was also interested in the rationale behind the length of the window used and the frequency bands. It wasn't until I had read the "Methods" section after reading the conclusion that those questions were answered. It would be good to either merge the information from both sections such that all of the information about the methodology is presented before the results, or to reference the "Methods" section from within the "Sampling Mercury's foreshock with MESSENGER MAG" section where appropriate. Thank you, the new version of the manuscript makes reference to 'Methods' in the ULF Wave detection sub-section. We have also adapted these sections in order to adhere to the formatting requirements of Nature Communications.

105: Are the perpendicular components combined prior to the FFT analysis? If so, would you run the risk of losing information using this method? If there is a circularly polarized oscillation with a constant amplitude in the perpendicular components, then surely that would manifest in a constant \mathbf{B}_\perp and not be picked up by the FFT? We thank the reviewer for this question. The Power Spectral Density of \mathbf{B}_\perp is the result of the combination of the FFT of both perpendicular components. In other words, the perpendicular components are combined in the Fourier space, not in the real space. By doing this, waves can be detected even if the amplitude of $|\mathbf{B}_\perp|$ is constant.

107: Please define what the sub-interval is. (i.e. is it some fraction of the 204.8 s interval? If so, how long are they?) Thank you, as the reviewer is stating, it is a fraction of the 204.8 s interval. We have defined the sub-interval size (14s - 50s) in line 366.

136-138: is it possible that this limit appears as a result of the wave selection process? As the reviewer is suggesting, the limit reported in this part of the manuscript is related to the wave selection criteria considered with Figure 1. We have chosen $r=2$ following a conservative approach since, in order to detect waves, the PSD at the wave frequency range of interest must be higher than the background solar wind turbulent spectrum. We have performed corresponding tests and we did not find a significant variability of this limit for the explored wave selection criteria. We have included a sentence in the manuscript assessing this point in Lines 136-139.

Figure 6b - While I like the look of this plot, how confident are you in the fit? A tiny part of me worries that the nice blue line might just be guiding my eye to believe that I see an 88-day pattern in the wave occurrence rate, when there are obviously other periods present. A good thing to show here would be a FFT/Lomb-Scargle analysis of this plot to show that there is a peak at the orbital period of Mercury. ? We thank the reviewer for this comment. We are confident in this fit, as the parameters are derived following a Least squares minimization approach. In addition and, following the reviewer suggestion, we have performed a FFT/Lomb Scargle analysis that shows the presence of a dominant peak at $T \sim 88.2$ days, consistent with the orbital period of Mercury. We have included an additional Figure (new Figure 7) in the manuscript showing this analysis and a description of the results is provided in Lines 198-201.

193-196: What is the phase shift/how many days out of phase? The phase shift is ~ 0.4 hr. We added this information in the manuscript.

314: It might be a good idea to link directly to the MAG data. Thank you, we have included the direct link to the MAG data.

Reviewer #2 (Remarks to the Author):

The manuscript is well written and presents interesting results about ULF waves in the Hermean environment, namely the foreshock. The conclusions are relevant for the space plasmas community as they give insight and enhance current understanding of how wave power varies with plasma conditions in shock related regions. The paper can be accepted for publication after some points described below have been clarified.

Line, 7. While planetary shock properties depend on solar wind parameters, the sentence “to elucidate the transfer of energy from the Sun to planetary systems“ is misleading on this line. Thank you, we have removed this part of the sentence.

Line 41, ULF waves are not “a type” of electromagnetic plasma waves, several modes can be generated in foreshock regions. Thank you, we have rewritten this sentence.

Line 54, “low Mach number” how low? The solar wind Alfvénic Mach number ranges between ~3 and ~6. We have included this information in the new version of the manuscript.

Line 105, Bcomp refers to parallel?, specify, why not use “paral”, or the parallel sign. Yes, Bcomp refers to parallel. We have replaced Bcomp by Bparallel.

Line 141, authors claim that ULF waves are not observed for a perpendicular shock because they do not have time and space to grow, this is misleading, ion escape from the shock is dramatically reduced after a certain value of oblique theta_Bn, so instability conditions for grow do not exist. Some words about variation of ion fluxes with shock geometry need to be included here and relate to conditions of instability, etc

Thank you, we have rewritten this sentence (Lines 139-144) in connection with the ion escape from the shock, providing also a brief mention to the expected variation of the ion fluxes with the shock geometry based on studies performed at the terrestrial foreshock. We have also included references to the following works, where the observed dependence at the Earth’s foreshock is reported in detail:

Bonifazi, C., and G. Moreno (1981), Reflected and diffuse ions backstreaming from the Earth's bow shock. 1: Basic properties, J. Geophys. Res., 86, 4397-4413.

G. Paschmann, N. Sckopke, I. Papamastorakis, J.R. Asbridge, S.J. Bame, J.T. Gosling, Characteristics of reflected and diffuse ions upstream from the earth’s bow shock. J. Geophys. Res. 86, 4355–4364 (1981). doi:10.1029/JA086iA06p04355

Burgess, D., E. Mobius, and M. Scholer (2012), Ion acceleration at the Earth’s bow shock, Space Sci. Rev., 173, 5–47, doi:10.1007/s11214-012-9901-5.

It is also worth mentioning that the upcoming magnetic field and plasma observations by Bepi-Colombo will be of utmost importance to properly characterize the Hermean ion foreshock based on charged particle measurements, and identify similarities and differences with other solar system planetary foreshocks.

Line 244, needs rewording, as it is, it seems that waves increase the Mach number, not the value of B. Thank you, we have rewritten this sentence.

Line 268, as the authors state, planetary bow shocks in the solar system have larger Alfvén Mach number than at Mercury. However, interplanetary shocks can have low Mach numbers and nothing is mentioned about them. A few lines related to these shocks and results present in this study are needed.

Thank you, we have added this information in Lines 315-322, together with corresponding references.

Minor

Line 14 Alfven Mach --- > Alfvén Mach number Corrected.

Reviewer #3 (Remarks to the Author):

This paper investigates the generation of fast magnetosonic plasma waves observed over four years with the MESSENGER magnetometer in the foreshock region of Mercury. The waves are in the 0.05-0.41 Hz frequency range and are most likely produced by cyclotron resonance between the solar wind plasma and backstreaming ions reflected from the Hermean bow shock. The authors show that these waves occur preferentially <5 RM upstream under quasi-radial conditions, and with lower IMF intensity and larger heliocentric distance. There is a suggestion some other variable also influences wave occurrence.

The authors have analysed a very large data set and drawn inferences which provide new insight on the generation of low frequency plasma waves in planetary foreshocks. This is both topical and timely work for planetary scientists. The manuscript is generally well written and the figures are well presented. However, I have some concerns regarding the scope of the data interpretation, and minor comments about typos, etc. These are listed below. The paper should be accepted for publication once these have been addressed.

Main points

1. The paper focuses on magnetosonic foreshock waves in the 0.05-0.41 Hz frequency range.

(a) Please explain or justify this choice of frequency. Thank you, we have included information on this matter in the Methods Section, Lines 356 – 362.

(b) Le et al., J. Geophys. Res. 118, 2013, found that these waves are relatively weak and occur only sporadically in Hermean foreshock (likely because of the low Alfvénic Mach number), while ~1 Hz whistler mode waves are much more abundant in the foreshock, sometimes appearing in wavetrains lasting hours. Figures 3(a), 4(a) and 6(a) in the present paper clearly show the occurrence of the magnetosonic waves is very low, but the manuscript is silent regarding the 1 Hz whistler waves. Please clarify the occurrence and significance of the two wave populations, especially in regard to the discussion and conclusions about waves in planetary foreshock regions.

Thank you, we have added this information in Lines 58-68 and in Lines 264-266. We would like to point out that although the results presented in Le et al 2013 suggest that 1Hz whistler waves do occur more frequently than 0.05-0.41 fast magnetosonic waves, an extensive analysis on the whistler wave population is needed to test if such trend is observed for a larger statistical data set. Indeed, Le et al. (2013) performed an analysis of a survey of waves observed during a single Hermean foreshock passage on 26 March 2011. Such statistical analysis is beyond the scope of the present study since the current manuscript is not focused on 1Hz whistler waves.

As the reviewer is stating, Le et al 2013 also found that the lowest-frequency waves were present sporadically in Mercury's foreshock, had small amplitudes ($\text{dB}/B_0 \sim 0.1$), and a frequency ~ 0.3 Hz. In this regard, the statistical results by Romanelli et al 2020 and those in the present manuscript are in agreement with the conclusions reported by Le et al 2013.

2. Data selection and analysis is based on comparison of peak power spectral density in the frequency range of interest with that in lower and higher frequency bands. The ratio of these values is given by r . For the figures presented in this paper $r=2$, while in Romanelli et al (2020) $r=4$. Figure 3(b) shows that wave occurrence with heliocentric distance is quite different for $r=10$ (a stricter selection criterion) than for $r=2$. An additional selection criterion is the ratio of intermediate and minimum eigenvalues of the covariance matrix of the magnetic field in the given time interval, λ_2/λ_3 ($=5$ in Romanelli et al 2020). Please justify this choice of selection criteria. For example, would there be any change in the wave occurrence rate with IMF intensity, shown in Figure 4(b), for higher r or different λ values?

We thank the reviewer for this comment. Firstly, we would like to mention that the point behind Figure 3b is to show there is an increasing trend between the wave occurrence rate and Mercury's heliocentric distance, for all the considered wave selection criteria. Although the wave occurrence rate is significantly smaller for $r=10$ (compared to the other curves), there is still an increasing trend (also smaller) for these relatively high amplitude ULF waves. As shown in Table 1, the R-value is 0.84 and the p-value is equal to $4.6e-3$, suggesting that the increasing linear trend is indeed significant. Differences in the R-values and the best linear fits are associated with the wave frequency and polarization properties and the time required to reach such states. Despite of these differences, the increasing trend between wave occurrence rate and Mercury's heliocentric distance is present regardless of the selection criteria considered in this paper.

As stated in Lines 174-175, the waves are observed preferably for a quasi-radial IMF, regardless of all explored selection criteria. In addition, as reported in Lines 171-172, a decreasing trend between the wave occurrence rate and $|B|$ (with a similar power law index to the one reported for Figure 4b) is obtained when the other selection criteria presented in Table 1 are applied.

The results presented in Romanelli et al (2020) correspond to $r = 4$, and $\lambda_{23} = 5$. However, the authors also stated that they did not find significant differences in the reported results when r is varied between 2 and 10. In this regard, it is important to emphasize that Romanelli et al 2020 was concerned with the frequency and polarization properties, that is, the direction of wave propagation and the normalized wave amplitude distributions, the relationship between the observed wave frequency and $|B|$, and the implications for the speed of backstreaming protons. In the current paper, we are concerned with the spatial distribution of the waves, the conditions under which they occur more frequently and the relationship between the wave occurrence rate and the heliocentric distance. Following the reviewer's suggestion, we have included a justification for the wave selection criteria in Lines 356-362, in addition, to the paragraph in Lines 370-379.

3. For Earth the magnetosonic waves generated in the foreshock have period around 30 sec and propagate into the magnetosheath and magnetosphere where they provide an importance source of wave energy. Please comment on what could be expected for Mercury, and indeed other planets with similar solar/stellar wind environments.

Thank you, we have provided information on some of the reported consequences of foreshock waves inside of several solar system planetary magnetospheres in Lines 255-269. Among them, we mention the propagation and energy transfer inside the magnetopause, the heating of the ionosphere and the modulation of planetary heavy ion escape fluxes. Although these processes suggest a similar coupling inside the Hermean magnetosphere, the relative importance of foreshock waves with respect to other frequent phenomena (e.g., magnetic reconnection) around Mercury, requires more analysis, ideally making use of multi-spacecraft measurements. The latter would allow to improve the current understanding on this topic, partly, by identifying a correlation between foreshock phenomena and observed signatures inside the magnetosphere of Mercury.

4. On page 14, lines 249, 250 is the statement "detection of the ULF waves throughout Mercury's orbit suggests the presence of a relatively stable backstreaming ion population." This is misleading for two reasons. First, the solar wind at Mercury is highly variable so would not produce a 'relatively stable'

backstreaming population. Second, the Mach number at Mercury is fairly small so the backstreaming ion beams are weak (compared with Earth's foreshock) and the resultant waves are much smaller than at Earth. Please amend the text and discussion accordingly. See e.g. Zurbuchen et al., Adv. Space Res., 33, 2004, and Le et al, 2013. We thank the reviewer for this comment. We meant that the detection of ULF waves throughout Mercury's highly eccentric orbit suggests that the conditions for backstreaming protons are potentially present for all Mercury's heliocentric distances, despite the relatively low SW Alfvén Mach number regime. We have amended the abstract and discussion accordingly, to bring clarity to the manuscript.

Other points

Page 2, line 48. "... with, after..." Word missing Corrected.

Page 5, line 115. "... our criteria state that" Corrected.

Page 6, line 123. "... 204.8 s interval" Corrected.

Page 6, line 129. "... Romanelli et al (2020), where $r=4$ and $\lambda_{2,3}=5$." Corrected.

Page 6, line 147. "... This allows us" Corrected.

Page 7. Figure 2 caption. "... The middle point in each bar corresponds..." Corrected.

Page 7, lines 151, 152. The statement that the trend for wave occurrence to increase with heliocentric distance regardless of the wave selection criteria is questionable. The trend decreases with increasing r and is hardly present for $r=10$.

We understand the reviewer's point of view. The slope associated with the wave occurrence rate as a function of heliocentric distance decreases for increasing values of the r parameter. However, as can be seen in Table 1 and the supplementary Excel file, the slopes associated with these fits are all positive. Thus, the increasing trend is present for all explored wave selection criteria.

Interestingly, Table 1 and Figure 3b show that the linear trend is present more clearly when we only consider the PSD properties (r varying between 2 and 10, with $\lambda_{23}=0$). Indeed, the inclusion of the wave polarization criteria reduces the linear correlation factor R from 0.89, 0.86, 0.84 (for $r=2, 4, 10$) to 0.76. Such decrease in R can be seen in Figure 3b, as the best linear fits for $r=2$ and $r=4$ present smaller slopes when the polarization properties are taken into account in the wave selection criteria. The best linear fit associated with $r=10$, presents an R -value very close to that of $r=2$ and $r=4$, and all of them present a p -value smaller than 0.05 suggesting the linear correlations are significant.

One possible explanation for such differences could be associated with the temporal evolution of the waves. Indeed, even though the most likely source of all observed fast magnetosonic waves are backstreaming ions, the waves that present a clear peak in the PSD and a well-defined polarization state (e.g., $r=2, \lambda_{23}=5$) are likely in an earlier evolution stage than some of the waves that only kept the peak in the PSD (e.g., $r=2, \lambda_{23}=0$) [see, e.g., Mazelle and Neubauer 1993, Romanelli et al., 2016 and references therein]. Given that $r=2, \lambda_{23}=5$ is a subset of $r=2, \lambda_{23}=0$, some waves events are included in both sets.

If this is the case, then the increase in the slope between wave occurrence rate as a function of Mercury's heliocentric distance when considering a fixed value of r , but reducing the value of λ_{23} , can be understood in terms of the time needed to reach a given wave stage. In other words, waves with $r=2$ and $\lambda_{23}=5$ allow us to identify waves in an earlier stage, while $r=2$ and $\lambda_{23}=0$ (being a set including the previous one) is

associated with both, early and more evolved waves. Given that the latter most likely require more time with the appropriate external conditions, the fits shown in Figure 3b suggest that the conditions favorable for these ULF waves occur more frequently as Mercury's heliocentric distance increases.

Finally, the fact that the wave occurrence rate is decreasing with the value of r can be understood in terms of the necessary energy input and time needed to reach the highest amplitude waves. Indeed, it is reasonable to consider that the wave occurrence rate should be zero for all heliocentric distances above a given threshold for r . Indeed, $r=10$ seems to be closer to the limit where there is not sufficient free energy and time for the development of such high amplitude waves. Figure 3b therefore shows a gradual decrease in the occurrence of waves satisfying $r=2$ and higher amplitude waves ($r=10$), that could be indicative of a limit to the maximum amplitude achievable in the Hermean foreshock for all Mercury's heliocentric distances.

We have included information on this matter in Lines 284-305, following this reviewer's comment.

Page 9, line 189. "... through all the MESSENGER mission" Corrected.

Page 13, line 225. "... for a fixed value" Corrected.

Page 14, lines 238, 239. "Wave are seen most of the time..." This statement could be misleading. Waves do occur preferably under these conditions but wave occurrence is never high. Corrected.

Page 14, line 243. "Waves are seen most of the time..." Same comment as above. Corrected.

REVIEWERS' COMMENTS

Reviewer #1 (Remarks to the Author):

The authors have addressed all of my comments and questions from the first review and I am satisfied with all of their responses. New additions to the paper look good and I am happy to recommend it for publication.

Reviewer #2 (Remarks to the Author):

The manuscript has improved after the revision and in my opinion can be accepted for publication. The material presented provides important information about Mercury's foreshock and about solar wind interaction with planets.

Reviewer #3 (Remarks to the Author):

I have reviewed the authors' responses to referees' reports, and the revised manuscript.

The authors have satisfactorily addressed my concerns and the paper should now be published. I have a few minor corrections, listed below, which can be attended to in the proof stage.

Page 2, line 26. "... environments such as "

Page 4, line 84. "...where 0.05-0.4 Hz ULF waves"

Page 6, line 125. "... in the Methods Section"

Page 10, line 191. "... IMF intensity throughout the MESSENGER mission

Page 11, line 195. "...correlated with the planet's"

Page 13, line 226. "Despite these ..."

Page 14, line 234. "... a scheme"

Page 17, line 311. "... allows us to identify"

Page 18, line 347. "We conclude that"

Page 18, line 353. "... allow us to compute"

This is our answer to the referee's reports of the article entitled: 'Occurrence rate of ultra-low frequency waves in Mercury's foreshock increases with heliocentric distance'.

Once again, we thank the reviewers for devoting their time to review our paper, and for their encouraging comments and useful suggestions. Their comments have helped us to improve the clarity and state of the manuscript. In the following, they can find our answers (in blue) inserted after each specific point. This version of the manuscript also presents modifications, to adhere to the formatting requirements of Nature Communications.

Yours sincerely,

Norberto Romanelli and Gina A. DiBraccio.

REVIEWERS' COMMENTS

Reviewer #1 (Remarks to the Author):

The authors have addressed all of my comments and questions from the first review and I am satisfied with all of their responses. New additions to the paper look good and I am happy to recommend it for publication.

Thank you.

Reviewer #2 (Remarks to the Author):

The manuscript has improved after the revision and in my opinion can be accepted for publication. The material presented provides important information about Mercury's foreshock and about solar wind interaction with planets.

Thank you.

Reviewer #3 (Remarks to the Author):

I have reviewed the authors' responses to referees' reports, and the revised manuscript.

The authors have satisfactorily addressed my concerns and the paper should now be published. I have a few minor corrections, listed below, which can be attended to in the proof stage.

Thank you.

Page 2, line 26. "... environments such as " **Corrected.**

Page 4, line 84. "...where 0.05-0.4 Hz ULF waves" **Corrected.**

Page 6, line 125. "... in the Methods Section" **This sentence was modified to comply with formatting requirements.**

Page 10, line 191. "... IMF intensity throughout the MESSENGER mission **Corrected.**

Page 11, line 195. "...correlated with the planet's" **Corrected.**

Page 13, line 226. "Despite these ..." **Corrected.**

Page 14, line 234. "... a scheme" **Corrected.**

Page 17, line 311. "... allows us to identify" Corrected.

Page 18, line 347. "We conclude that" Corrected.

Page 18, line 353. "... allow us to compute" Corrected.